# Chemical potential of quasi-equilibrium magnon gas driven by pure spin current

V.E. Demidov[1], S. Urazhdin[2], B. Divinskiy[1], V.D. Bessonov[3], A.B. Rinkevich[3], V.V. Ustinov[3,4] & S.O. Demokritov[1,3]

Pure spin currents provide the possibility to control the magnetization state of conducting and insulating magnetic materials. They allow one to increase or reduce the density of magnons, and achieve coherent dynamic states of magnetization reminiscent of the Bose–Einstein condensation. However, until now there was no direct evidence that the state of the magnon gas subjected to spin current can be treated thermodynamically. Here, we show experimentally that the spin current generated by the spin-Hall effect drives the magnon gas into a quasi-equilibrium state that can be described by the Bose–Einstein statistics. The magnon population function is characterized either by an increased effective chemical potential or by a reduced effective temperature, depending on the spin current polarization. In the former case, the chemical potential can closely approach, at large driving currents, the lowest-energy magnon state, indicating the possibility of spin current-driven Bose–Einstein condensation.

[1] Institute for Applied Physics and Center for Nonlinear Science, University of Muenster, Corrensstrasse 2-4, 48149 Muenster, Germany. [2] Department of Physics, Emory University, Atlanta, GA 30322, USA. [3] Institute of Metal Physics, Ural Division of RAS, Ekaterinburg 620108, Russia. [4] Institute of Natural Sciences, Ural Federal University, Ekaterinburg 620083, Russia. Correspondence and requests for materials should be addressed to V.E.D. (email: demidov@uni-muenster.de)

The discovery of the room temperature magnon Bose–Einstein condensate (BEC) in magnetic insulators driven by parametric pumping[1] has spurred intense experimental and theoretical studies of this phenomenon[2–11]. It is now well-established that the chemical potential of the magnon gas is increased by the parametric pumping, resulting in the formation of BEC when it reaches the lowest-energy magnon state[1]. The formation of magnon BEC has been experimentally confirmed by the observation of the spontaneous narrowing of the population function in the energy[3,11] and phase space[4]. Moreover, phase coherence of this state has been confirmed by the observation of interference in the real space[6].

Although parametric driving provides a convenient approach to studies of BEC, it also has shortcomings. In particular, the energy of magnons injected by the parametric pumping is concentrated within a narrow range[12], initially producing a strongly non-thermal state of the magnon gas. A significant thermalization time is required before a quasi-equilibrium state with non-zero chemical potential is formed[2,3]. The accompanying increase of the effective temperature of low-energy magnons can be also detrimental to the formation of BEC[7].

The magnon gas can be driven instead by the injection of spin current[13] generated, for instance, by the spin-Hall effect (SHE)[14–16]. As was shown in ref.[13], injection of spin current results in either enhancement or suppression of magnetic fluctuations, depending on the polarization, which can be equivalently described as generation or annihilation of incoherent magnons. This mechanism is not specific to certain magnon states, and is expected to change magnon populations throughout the entire spectrum, thus avoiding the non-thermalized transient states inherent to parametric driving. Recent theoretical studies[17–21] suggest that spin current can drive the magnon gas into a quasi-equilibrium state described by the Bose–Einstein statistics with non-zero chemical potential, suggesting the possibility of BEC formation at sufficiently large currents. These theories have been supported by the successful application of the developed theoretical framework to incoherent magnon transport[20,22]. Variations of the chemical potential of the magnon gas were recently detected in measurements of spin relaxation rates of a nitrogen-vacancy center in diamond coupled to spin waves in a magnetic insulator[23]. However, there is no direct experimental evidence that the magnon gas driven by pure spin current forms a quasi-equilibrium distribution, and the dependence of the effective thermodynamic characteristics on spin current has not been established.

Here, we utilize a Permalloy/Pt bilayer to study the effect of pure spin current on the magnon distribution over a significant spectral range, allowing us to demonstrate that this distribution can be described by the Bose–Einstein statistics expected for the quasi-equilibrium state, and determine the current-dependent chemical potential and effective temperature. We show that, for one polarization of the spin current, the effective temperature of the magnon gas becomes significantly reduced, whereas the chemical potential stays almost constant. In contrast, for the opposite polarization, the effective temperature remains nearly unaffected, whereas the chemical potential linearly increases with current until it closely approaches the lowest-energy magnon state.

## Results

### Studied system and experimental approach.
The system comprises a 2 μm wide and 5 nm-thick Pt strip overlaid by a 1 μm wide and 10 nm-thick ferromagnetic Permalloy (Py) strip (Fig. 1a). The independently measured saturation magnetization of Py is $4\pi M_0 = 10.2$ kG. The system is magnetized by the

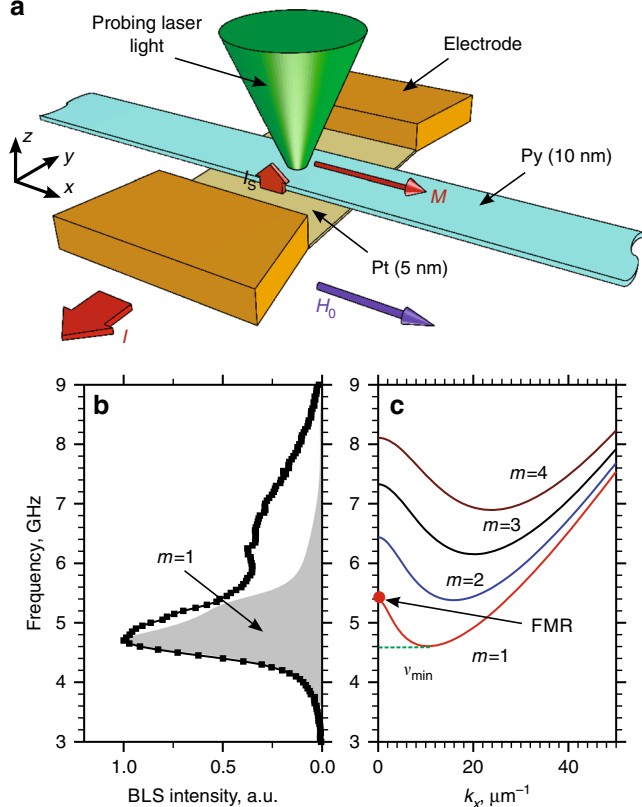

**Fig. 1** Studied system. **a** Schematic of the experiment. **b** BLS spectrum of magnons in the Py strip measured at $I = 0$ and $H_0 = 200$ Oe. Shaded area shows the calculated BLS response for the fundamental magnon mode $m = 1$. **c** Calculated dispersion spectrum of magnon modes in the Py strip. $m$ is the mode index, $\nu_{min}$ is the frequency of the lowest-energy magnon state

static magnetic field $H_0$ applied along the Py strip. For the studied 15 μm-long strip, the inhomogeneous dipolar field is negligible in the active device area. The electric current $I$ flowing in Pt is converted by SHE into a spin current $I_S$ injected into Py through the Py/Pt interface. The magnetic moment carried by the spin current is either parallel or antiparallel to the Py magnetization $M$, depending on the direction of current[24], resulting in a decrease or an increase of the magnon population, respectively[13].

We study the magnon population by the microfocus Brillouin light scattering (BLS) technique[25]. We focus the single-frequency probing laser light with the wavelength of 532 nm onto the surface of the Py strip, and analyze the light inelastically scattered from magnons. The measured signal—the BLS intensity—is directly proportional to the spectral density of magnons $\rho(\nu) = D(\nu)n(\nu)$, where $\nu$ is the magnon frequency, $D(\nu)$ is the density of magnon states weighted by the wavevector-dependent measurement sensitivity, and $n(\nu)$ is the occupation function[1].

A representative BLS spectrum recorded at $H_0 = 200$ Oe and $I = 0$ exhibits a peak with the highest intensity in the frequency range $\nu = 4$–5.5 GHz, and a shallow high-frequency tail extending to 9 GHz (Fig. 1b). The origin of these spectral features is elucidated by the analysis of magnon dispersion in the Py strip (Fig. 1c), which is calculated using the approach described in ref.[25]. The spectrum is quantized in the direction transverse to the Py strip, and is continuous in the longitudinal direction. The allowed transverse wavevector components are $k_y \approx \pi m/w$, where $w$ is the width of the Py strip, and positive integer $m$ is the mode

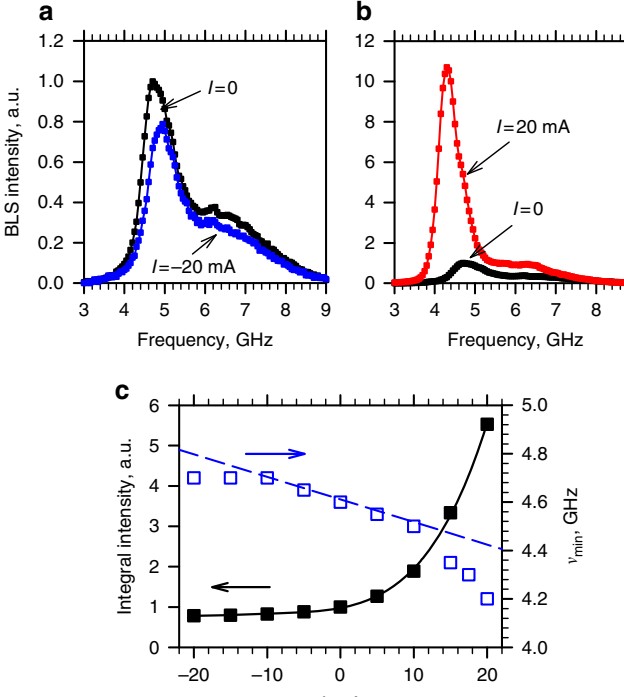

**Fig. 2** Effects of spin current. **a, b** Representative BLS spectra recorded at $I$ = −20 and 20 mA, respectively, together with the reference spectrum obtained at $I = 0$. **c** Current dependences of the BLS intensity integrated over the measured spectrum (solid squares) and of the frequency of the lowest-energy magnon state $\nu_{min}$ (open squares). Solid curve is a guide for the eye. Dashed line shows the calculated variation of $\nu_{min}$ due to the Oersted field of the current. The data were obtained at $H_0 = 200$ Oe

index. Because of the dependence of the BLS sensitivity on the wavevector, the fundamental mode $m = 1$ provides the largest contribution to the BLS spectrum, producing the peak observed at $\nu = 4$–5.5 GHz, as indicated in Fig. 1b by the shaded area. Despite the reduced BLS sensitivity to the higher-order modes, it allows measurements of the magnon population in a broad range of frequencies up to $\nu = 9$ GHz. Our measurements also provide an opportunity to distinguish between the quasi-uniform ferromagnetic resonance (FMR) mode with $k_x = 0$ and the lowest-energy finite-wavevector magnon state at frequency $\nu_{min}$ (Fig. 1c), as their frequency separation of about 0.8 GHz is significantly larger than the resolution of the BLS. This distinguishing feature allows us to determine which of these states is predominantly overpopulated due to the spin current injection, addressing a long-standing debate in the studies of spin current-induced effects[26].

**Effects of the spin current on the magnon gas**. To reduce the effects of Joule heating on the current-dependent BLS spectra, in our measurements the current was applied in 200 ns-long pulses with the repetition period of 1 µs, and the BLS spectra were accumulated over the duration of the pulse. The effects of current are illustrated in Fig. 2a, b for $I = −20$ and 20 mA, respectively. At $I < 0$, the BLS intensity decreases, and the spectrum shifts to higher frequencies. In contrast, at $I > 0$ the BLS intensity strongly increases, whereas the spectrum shifts to lower frequencies.

Solid squares in Fig. 2c show the current dependence of the intensity integrated over the measured spectrum, which characterizes the total number of low-energy magnons accessible to BLS. This dependence is consistent with the expected

reduction/increase of the magnon population by the appropriate polarization of spin current[13].

The dependence of the lowest magnon frequency $\nu_{min}$ on current (open squares in Fig. 2c) can be attributed to a combination of the Oersted field of the current and the variation of the effective magnetization of Py due to the effect of spin current on the magnon population, as well as Joule heating of the sample. The calculated contribution of the Oersted field is shown in Fig. 2c by the dashed curve. The experimental data closely follow this dependence at $|I| < 10$ mA, and deviate from it at larger current magnitudes. The deviation is larger at $I > 0$ than at $I < 0$. As Joule heating does not depend on the sign of current, we conclude that the total magnon population that determines the effective magnetization is significantly affected by the spin current.

We now analyze the spectral distribution of spin current-driven magnon population, by comparing the zero-current BLS spectra with those obtained at finite currents. At $I = 0$, the magnon gas is in thermal equilibrium, with the temperature equal to the experimental temperature $T_0 = 295$ K, and the chemical potential $\mu = 0$[1]. Correspondingly, the measured BLS intensity is proportional to the weighted spectral density of magnons $\rho_0(\nu) = D(\nu)n_0(\nu)$, where $n_0(\nu)$ is the Bose–Einstein distribution. In the limit $h\nu \ll k_B T_0$, the latter is well approximated by the Rayleigh–Jeans law $n_0(\nu) = k_B T_0 / h\nu$, where $k_B$ is the Boltzmann constant and $h$ is the Planck constant. At finite spin current, $\rho_I(\nu) = D(\nu)n_I(\nu)$ with a current-dependent distribution $n_I(\nu)$. If the magnon gas is driven into a quasi-equilibrium state, this distribution can be written as $n_I(\nu) = k_B T_{eff} / (h\nu - \mu)$, with effective temperature $T_{eff}$ and chemical potential $\mu$. The density of states $D(\nu)$ is not expected to be influenced by the spin current, aside from the frequency shift discussed above. Therefore, the ratio of the BLS signals measured with and without current, or equivalently the frequency-dependent enhancement of the magnon population, is

$$R(\nu) = \frac{T_{eff}}{T_0} \frac{\nu}{\nu - \mu/h}. \qquad (1)$$

This relation allows us to test whether the current-dependent magnon populations are well described by the quasi-equilibrium distribution, and extract the current-dependent values of $T_{eff}$ and $\mu$. Note that the roles of these parameters in Eq. (1) are qualitatively different: variations of $T_{eff}$ result in frequency-independent scaling of $R$, whereas $\mu > 0$ produces a monotonically decreasing dependence $R(\nu)$ approaching 1 at large $\nu$.

Solid symbols in Fig. 3a, b show on the log-linear scale the BLS spectra recorded at $I = −20$ and 20 mA, respectively. Open symbols in the same plots show the spectrum obtained at $I = 0$, shifted in frequency by the value determined from the data in Fig. 2c. The data in Fig. 3a illustrate that at $I < 0$, the magnon populations decrease approximately uniformly over the entire frequency range of the detected spectrum (see also Supplementary Fig. 1). In contrast, the increase of the population at $I > 0$ (Fig. 3b) is most significant at the frequency $\nu_{min}$ of the lowest-energy magnon state. The increase is smaller by more than a factor of two at the frequency of the quasi-uniform FMR mode. It is further rapidly reduced at higher frequencies, suggesting that the effects of spin current at $I > 0$ are qualitatively different from those at $I < 0$.

Figure 3c shows the ratio of the spectra obtained with and without current. For $I = −20$ mA (open symbols in Fig. 3c), this ratio is independent of frequency. According to Eq. (1), this indicates that the dominant effect of spin current at $I < 0$ is the reduction of the effective temperature, $T_{eff} \approx 0.76 T_0 = 224$ K at $I = −20$ mA. The frequency-dependent enhancement of the magnon population at $I = 20$ mA (solid symbols in Fig. 3c) is also

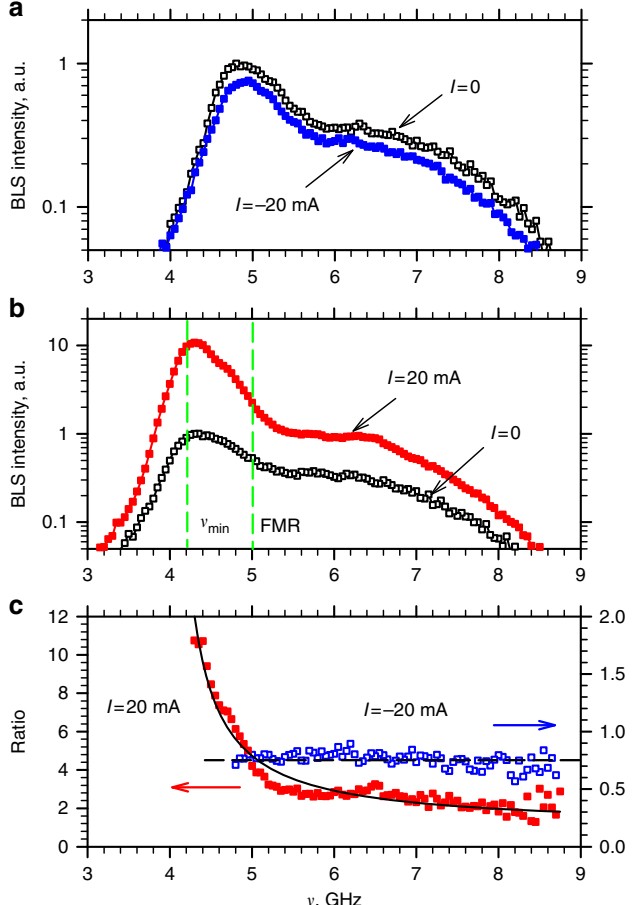

**Fig. 3** Spectral distribution of spin current-driven magnon population. **a**, **b** BLS spectra recorded at $I = -20$ and 20 mA, respectively, (solid symbols) together with the reference spectrum obtained at $I = 0$ (open symbols) shifted in frequency by the value determined from the data of Fig. 2c. Vertical dashed lines in **b** mark the frequency $\nu_{min}$ of the lowest-energy magnon state, and that of the quasi-uniform FMR mode. Note the logarithmic scale on the vertical axis. **c** The ratio of the spectra obtained with and without current. Dashed line is the mean value of the ratio for $I = -20$ mA. Solid curve is the fit of the ratio for $I = 20$ mA by Eq. (1), with $T = T_0$ and $\mu/h = 3.94$ GHz

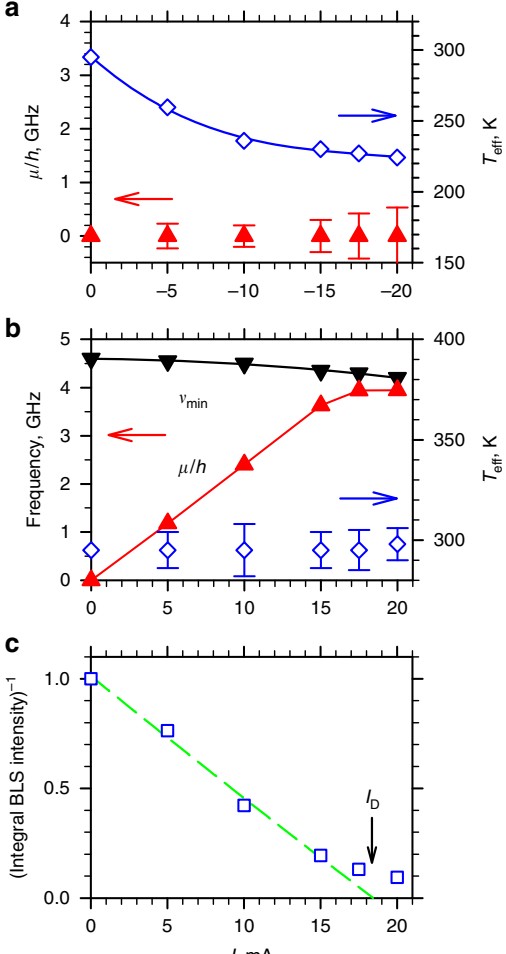

**Fig. 4** Thermodynamic characteristics of the quasi-equilibrium magnon gas. **a**, **b** Current dependences of $\mu/h$ (point-up triangles) and of the effective temperature (diamonds) of the magnon gas for $I < 0$ (**a**) and $I > 0$ (**b**). Point-down triangles in **b** show the frequency of the lowest-energy magnon state $\nu_{min}$. Curves are guides for the eye. Error bars show the fitting uncertainty (s.e.m.). For clarity, the error bars are only shown if the error exceeds the size of the symbols. **c** Current dependence of the inverse of the BLS intensity integrated over a 100 MHz window around $\nu_{min}$. Dashed line is the linear fit at small currents, with $I_D$ marking its intercept

well described by Eq. (1). In this case, a good fit is achieved with $T_{eff} = 298 \pm 8$ K, and the effective chemical potential in the frequency units $\mu/h = 3.94 \pm 0.02$ GHz (solid curve in Fig. 3c).

The validity of our analysis was confirmed by separate measurements of the dependence of BLS spectra on the sample temperature, at $I = 0$ (Supplementary Fig. 2). As expected, the BLS intensity simply scales by the frequency-independent ratio $T/T_0$.

### Discussion

Figure 4 summarizes the results of the same analysis performed for different currents. At $I < 0$ (Fig. 4a), the effective temperature monotonically decreases with increasing magnitude of $I < 0$, whereas the effective chemical potential remains zero within the measurement error. The effective temperature gradually saturates at large currents, which can be attributed to Joule heating that competes with the effects of spin current. Indeed, heat flow simulations show that the average increase of temperature in Py during the current pulse is about 45 K at $I = 20$ mA, comparable to the temperature reduction induced by the spin current. At $I > 0$ (Fig. 4b), the effective chemical potential increases linearly up to $I$

$= 15$ mA, reaching 80% of $h\nu_{min}$ at this current, whereas the effective temperature remains approximately equal to $T_0$. We note that the increase of temperature due to the Joule heating provides only a minor contribution to the magnon distribution, with the latter determined mostly by the increased chemical potential.

We emphasize that the dominance of the chemical potential increase at $I > 0$ does not imply that the effective temperature remains exactly equal to $T_0$. Conversely, the chemical potential may not remain exactly zero at $I < 0$. However, the effect of spin current on the effective temperature at $I > 0$, or on the chemical potential at $I < 0$ is too small to be reliably determined from the experimental data, as indicated by the error bars in Fig. 4a, b. On the basis of the general arguments of continuity, both parameters are expected to vary smoothly in the vicinity of $I = 0$. For instance, one can expect that the chemical potential becomes slightly negative at small $I < 0$, whereas the effective temperature slightly increases at small $I > 0$. Analysis of the data for small currents (see Supplementary Fig. 3) shows that the extrapolation of the linear dependence of the chemical potential, observed at $I > 0$, to $I < 0$ does not provide a satisfactory description of the

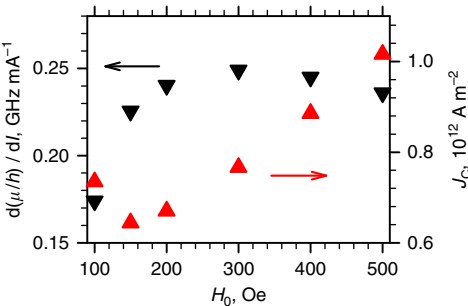

**Fig. 5** Dependence on the magnetic field. Static-field dependences of d($\mu/h$)/d$I$, the efficiency of spin current-driven chemical potential variation in frequency units (point-down triangles), and of $J_C$, the critical current density in Pt at which the chemical potential is expected to reach the energy of the lowest magnon state (point-up triangles)

experimental data even at a modest $I = -5$mA, indicating that the chemical potential quickly saturates at $I < 0$.

Extrapolating the linear dependence of the chemical potential (Fig. 4b) to larger currents, one would expect that it reaches the energy of the lowest magnon state at $I = I_C \approx 17.5$ mA, which should result in the formation of the BEC of magnons. Instead, the growth of $\mu$ rapidly saturates at $I > 15$ mA. This result is consistent with the previous studies, which showed that single-frequency current-driven magnetization dynamics cannot be achieved by injection of spin current into an extended region of the magnetic film[26], due to the onset of nonlinear magnon interactions in the strongly driven magnon gas that suppress the population of low-energy magnon states. It may be possible to overcome these effects by utilizing frequency-dependent magnon radiation losses[27], which are analogous to the evaporative cooling used in the experiments with atomic condensates[28]. However, this approach can significantly influence the quasi-equilibrium state of the magnon gas. Thus, one cannot unambiguously conclude that the previously observed coherent magnetization dynamics driven by spin currents[27,29–33] can be treated as the magnon BEC. Note that an alternative mechanism for the auto-oscillations is lasing, which results in the formation of a coherent state not associated with thermodynamic processes within the quasiparticle system, and generally cannot be described in terms of the effective temperature and chemical potential. According to the established criteria for BEC of quasi-particles[34], to be able to claim the current-driven BEC of magnons and to clearly distinguish it from lasing, one would need to demonstrate both the spontaneous emergence of coherence, and simultaneously the quasi-equilibrium state described by the Bose–Einstein statistics. Our results clearly indicate the feasibility of such a challenging demonstration. We also note that, as was shown in ref. [7], BEC can occur even in the absence of complete thermalization between low- and high-energy magnons, because of the weak coupling between these two groups. Therefore, evidence for the quasi-equilibrium state of the low-energy magnons, such as provided by our data, should be sufficient to demonstrate spin current-driven magnon BEC.

To elucidate the relation between the current-induced variation of the effective chemical potential and the current-induced auto-oscillations, we compare the critical current $I_C$ determined from the condition $\mu/h = \nu_{min}$ (Fig. 4b) with the current $I_D$ corresponding to the onset of dynamical instability associated with the complete damping compensation of the lowest-frequency mode. The value of $I_D$ was determined by analyzing the BLS intensity integrated over a 100 MHz window around $\nu_{min}$. The inverse of this quantity linearly depends on current (Fig. 4c), as expected for

the effects of spin current[26], with the extrapolated intercept at $I_D \approx 18$ mA close to $I_C \approx 17.5$ mA.

Next, we analyze the effects of the static magnetic field $H_0$ on the spin current-driven variations of the effective chemical potential. Measurements similar to those discussed above were performed at fields ranging between 100 and 500 Oe. Although the observed behaviors remained similar over the entire field range, the efficiency of the chemical potential variation by the spin current strongly depended on field. Since the dependence $\mu(I)$ is linear at moderate $I > 0$ (Fig. 4b), the spin-current efficiency can be characterized by the slope d($\mu/h$)/d$I$, as shown by the point-down triangles in Fig. 5. It rapidly increases with increasing small field, plateaus at $H_0 = 300$ Oe, and gradually decreases at larger fields. By extrapolating the linear dependence $\mu(I)$, we determine the critical electrical current density $J_C$ in Pt, at which the chemical potential would reach the energy of the lowest magnon state in the absence of the nonlinear suppression of magnon population (point-up triangles in Fig. 5). This dependence reaches a minimum at $H_0 = 150$ Oe, and linearly increases at larger fields. A similar dependence has been observed for the critical current in spin-Hall nano-oscillators[27,32]. We note that the auto-oscillation onset current densities in the latter are very close to the values of $J_C$ extrapolated from our measurements, confirming a close relation between the current-induced variation of the effective chemical potential and the current-induced auto-oscillations.

Finally, we discuss the nature of magnon pumping by spin current. We emphasize that the effects of spin current on the magnon gas cannot be interpreted as a broadband input of magnons. Instead, according to the well-established models based on the Landau–Lifshitz–Gilbert (LLG) equation with the Slonczewski's anti-damping torque[35], the flow of the angular momentum provided by the spin current is converted into magnons by the spin system of the ferromagnet. Therefore, magnon generation by the spin current is not a linear process, but rather involves feedback between the excitation source and the dynamics of the system. We note that to describe the fluctuation enhancement by the spin current, the LLG-based model must be augmented with an additional source of fluctuations, whose thermodynamic characteristics largely determine the resulting magnon distribution modulated by the spin current[26]. Thus, theoretical understanding of spin current-induced phenomena may be advanced by combining the widely used LLG-based approach with the thermodynamic description. We believe that our experimental study of the magnon distribution under the influence of spin current should provide a foundation for such theoretical studies.

In conclusion, our experimental results provide direct spectroscopic evidence that the magnon gas is driven by the pure spin current into a quasi-equilibrium state, which can be described by the Bose–Einstein distribution with current-dependent values of chemical potential and effective temperature. Our findings provide support for the theoretically proposed mechanism for formation of current-induced magnetization auto-oscillations via the Bose–Einstein condensation of magnons[17–19]. In contrast, lasing, also proposed as a possible mechanism for auto-oscillations, is not associated with the thermodynamic processes, and generally cannot be described by the effective temperature and chemical potential. BEC can be realized by avoiding the nonlinear magnon interactions that suppress the low-frequency mode populations at large magnon densities. Our results should stimulate further experimental and theoretical exploration of the relationship between the thermodynamics of magnon gases driven by spin currents and coherent magnetization dynamics.

## Methods

**Sample fabrication.** The studied structures were fabricated on annealed sapphire substrates with pre-patterned electrodes. First, a 2 μm-wide and 5 nm-thick Pt strip was fabricated by a combination of e-beam lithography and ultrahigh-vacuum sputtering at room temperature. Next, a 15 μm-long, 1 μm-wide, and 10 nm-thick $Ni_{80}Fe_{20}$ = Permalloy (Py) layer was sputtered on top of the Pt strip, and coated with a 5 nm-thick protective $SiO_2$ layer without breaking the vacuum. The Py strip was centered on the Pt strip, with its long direction perpendicular to the direction of the Pt strip. Finally, the entire structure was coated by a 50 nm-thick $SiO_2$ layer to prevent oxidation.

**Sample design.** To enable the study of the thermodynamic characteristics of the magnon gas under the influence of spin current, the design of our experimental system was substantially different from those utilized in the previous works on the excitation of magnetization dynamics by SHE[26,27,30–33]. To determine the effective temperature and chemical potential, one needs to experimentally access the spectral distribution of magnons over the range of frequencies comparable to the frequency of the lowest magnon state. This could not be realized in devices optimized for efficient excitation of auto-oscillations, which were based on 3–5 nm-thick active magnetic layers. The magnon dispersion is very flat in such thin films, resulting in a narrow frequency range accessible to BLS, comparable to the spectral resolution of the technique. Moreover, because of the spectral degeneracy, such devices did not allow one to experimentally distinguish the quasi-uniform magnon mode from the lowest-energy mode, which is necessary to distinguish BEC from the lasing regime. Finally, in all spin-Hall oscillator devices studied up to now, the static configuration of the magnetization was not spatially uniform due to the geometrical effects and the non-uniform Oersted fields of the driving current, resulting in a complex dispersion spectrum that strongly varied with current. These features make the previously studied systems unsuitable for the analysis of the magnon populations under the influence of spin current. To overcome these difficulties, we base our experimental system on a Py film whose relatively large thickness of 10 nm was chosen to satisfy two criteria. First, it results in a sufficiently steep dispersion spectrum of magnons, providing experimental access to the magnon states over a sufficiently broad range frequencies. Second, the effects of the spin current, which scale inversely with the film thickness, remain sufficiently large to analyze the effects at chemical potential approaching the lowest-energy magnon state. The Py film is shaped as a narrow wire with the magnetization directed along its axis. The quantization of magnon spectrum in the transverse direction removes the frequency degeneracy of the low-energy magnon modes, enabling us to experimentally distinguish quasi-uniform mode from the lowest-frequency mode. The large aspect ratio of the Py wire ensures that the internal static magnetic field and the static magnetization are uniform in the active device area, avoiding magnon edge states that could complicate the magnon spectrum, and allowing measurements at moderate static fields at which the accessible spectral range is comparable to the frequency of the lowest magnon state. The geometry of the sample also produces a uniform Oersted field of the driving current in the active device area, resulting in a negligible current-induced distortion of the magnon spectrum.

**Microfocus BLS measurements.** All the measurements were performed at room temperature. The probing light with the wavelength of 532 nm produced by a single-frequency laser was focused into a diffraction-limited spot on the surface of the Py film by using a ×100 microscope objective lens. The light inelastically scattered from magnons was collected by the same objective lens and analyzed by a six-pass Fabry–Perot interferometer. During the measurements, the position of the probing spot was kept constant with the precision better than 50 nm by using active stabilization.

**Calculations of the current and the heat flow.** The calculations were performed by using COMSOL Multiphysics simulation software (https://www.comsol.com/comsol-multiphysics). The independently measured thickness-corrected resistivities of the Pt and Py films 275 and 325 nm, respectively, were used in the calculations. The calculations showed that 35% of the total current flows in the Pt layer under the Py strip, producing magnetic field of 1.1 Oe mA$^{-1}$ in the latter. The shunting of the current through Py can, in principle, result in the modification of the dispersion spectrum of magnons via, for instance, the Doppler effect[36]. Estimates show that this effect results in a frequency shift of 10 MHz for the largest applied current magnitude of 20 mA, which is negligible compared to the characteristic frequency scale in our measurements. This is confirmed by the absence of noticeable modifications of the shape of the BLS spectra at $I = −20$ mA (Supplementary Fig. 1).

**Data availability.** The data that support the findings of this study are available from the corresponding author upon reasonable request.

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

## Acknowledgements

This work was supported by the Deutsche Forschungsgemeinschaft, the NSF Grant Nos. ECCS-1509794 and DMR-1504449, and the program Megagrant No. 14.Z50.31.0025 of the Russian Ministry of Education and Science.

## Author contributions

V.E.D. and B.D. performed measurements and data analysis. S.U. designed and fabricated the samples, and performed data analysis. V.D.B., A.B.R., and V.V.U. performed data analysis. S.O.D. formulated the experimental approach and managed the project. All authors co-wrote the manuscript.

## Additional information

**Competing interests:** The authors declare no competing financial interests.

