## [Peer Review File · Nature Communications]

Reviewers' comments:

Reviewer #1 (Remarks to the Author):

In this paper, Demidov et al. present a new interpretation of spin Hall driven magnetization dynamics in permalloy-platinum bilayers. The interest here is whether a Bose-Einstein condensate (BEC) can be formed via spin Hall driven pumping, rather than the parametric pumping previously demonstrated. While a BEC is not formed in this work, the analysis in terms of effective temperature and chemical potential indicate that this is possibility.

This is an interesting idea. However, I am unsure whether enough new material is presented here. The paper seems to be more about reinterpreting results that have been published previously, which were described using different language, see e.g. [PRL 107, 107204 (2011)].

An important aspect is the connection between auto-oscillations and BEC. However, the sentence “Our findings provide support for the theoretically proposed interpretation of current-induced magnetization auto-oscillations in terms of the BEC of magnons” does not have a citation. The only evidence given is that the critical currents are about the same. If this is a key result of the paper, then more convincing arguments should be presented. For example, the authors have elsewhere reached auto-oscillation in similar devices, does that data agree with the new interpretation? This is important as, if at some later stage someone was to publish a paper saying that they have achieved BEC of spin Hall driven magnons, would that be a) an interesting result or b) a trivial one because it has been observed previously, but was described as auto-oscillations of the magnetization? This is not clear to me from reading the paper, and it is therefore difficult to evaluate its importance.

Usually the anti-damping torque is written something like $M_x(M_x\sigma)$, in such a way that the magnetization effects the torque on itself. In this case, there is feedback of the magnetization dynamics into how the system is pumped. How does this usual form of the torque fit with the thermodynamic picture presented here, in which the pumping is consider to be simply broadband input of magnons? i.e. if there were coherent oscillations, would they be spontaneous due to condensation of magnons, or just from a feedback mechanism in the pumping?

In Fig. 3 (c), the scale on the negative current data is confusing because it is labelled enhancement, even though it is a suppression for negative currents. Would it not make more sense just to label the y-axis as the ratio defined in the text?

In Fig. 4 the temperature/chemical potential are not plotted over the entire current range. If the fits to the ratio using the expression in the text give meaningful values, then they should be presented.

In conclusion, in its current form I would not recommend this paper for publication in Nature Communications. The claims that are made are ambiguous, and the paper essentially presents the reinterpretation of previously demonstrated experimental results. While this interpretation is interesting, it is a small step towards the more important results of either a) demonstrating spin current driven BEC or b) showing convincingly that auto-oscillations previously observed correspond to BEC.

Reviewer #2 (Remarks to the Author):

The manuscript by Dr. Demidov et al. presents BLS measurements on a Py/Pt nanostructure while a current is driven through the system.

The main claims of the paper are that i) the spin Hall effect in Pt injects a spin current into Py which affects the magnon population, and ii) the magnon population is affected in a way that is described by a Bose-Einstein distribution function with nonzero chemical potential and an effective temperature.

These main claims are novel in the sense that the BLS measurements while driving current, performed by the authors, present new results. Previous work (including in part that of the authors themselves) has demonstrated, either directly or indirectly, the existence of nonzero chemical potential, either by injecting spin current or by driving the system in other ways. The authors refer appropriately to the experimental achievements in this direction, although some more theory work in this direction could also be cited.

The paper will surely be of interest to researchers working in this field: this work presents a novel experiment that indicates that the magnon chemical potential is a sensible quantity in describing a quasi-equilibrium magnon system. The field of magnon spintronics/spin caloritronics/spintronics is recently shifting away from electron spin currents to magnon spin currents. The work by Demidov et al. provides important input on how to describe the latter spin currents. As mentioned above, it does not stand alone in the sense that there are now several indications that the magnon chemical potential is a useful quantity. Together with these other works (e.g. Refs.17,21 of the manuscript) I expect that this paper will be very influential.

Regarding the validity of the claims, I have some concerns. That is, the claim rests on interpretation of the experimental data within the assumption that the main influence of the current is through the spin current in Pt. There are some aspects of this experimental system (i.e. a metallic magnet rather than an insulating one) and some aspects of the data that warrant questions:

*) To me, one of the most puzzling/intriguing aspects of the data is the asymmetry between the two directions of current. The authors do not discuss possible reasons for this in detail. In the linear response regime, one would expect effective temperature and chemical potential to be linear in the applied current, but I did not find an analysis that this is indeed the case – have the authors checked this? Could the authors provide more discussion of the asymmetry?

*) Since Py is a metal, there will be current flowing in it as well, and not only in the Pt – in fact, likely most of the current flows in the Py. Since the spin waves are not homogeneous in the direction of current, the current will influence the spin wave dispersion via the ordinary spin transfer torques (giving rise to what people call the spin wave Doppler effect). These will lower the $m=1$ mode for one direction of current, and raise it for the other one. Given the large current densities, it is not obvious to me that these effects may be ignored – have the authors included this effect in their analysis? Have the authors done a reference experiment without the Pt, e.g. replacing it with Cu?

*) If one estimates the change in total magnon density by estimating the injected spin and balancing it with relaxation, how does this then compare to the changes in the BLS data?

*) Are there Dzyalonskii-Moriya interactions induced by the Py/Pt interface?

*) The authors claim that the magnon gas is described by a Bose-Einstein distribution function with a nonzero chemical potential; strictly speaking they have only shown this for the low-energy tail of the distribution function.

*) In line 91,92 the authors mention a long-standing debate – have they now resolved it?

The main criterion for publication for Nature Communications is that “In general, to be acceptable, a paper should represent an advance in understanding likely to influence thinking in the field.” Provided that the authors answer my questions above in a satisfactory way I believe that this paper will become part of a few works that demonstrate that the magnon chemical potential is important in describing magnon systems driven by spin current and as such will indeed influence the thinking of the field.

Reviewer #3 (Remarks to the Author):

The manuscript by Demidov and co-workers addresses the phenomena through which a spin current may affect the magnon thermodynamics in a ferromagnetic material. The subject is novel, timely, and potentially interesting for the wide scientific community dedicated to Magnetism. The quality of the experimental data is very good and the manuscript provides strong evidence for its conclusions.

To my opinion, the manuscript is fit for publication in Nature Communications

Reply to Reviewer #1

The Reviewer writes:

In this paper, Demidov et al. present a new interpretation of spin Hall driven magnetization dynamics in permalloy-platinum bilayers. The interest here is whether a Bose-Einstein condensate (BEC) can be formed via spin Hall driven pumping, rather than the parametric pumping previously demonstrated. While a BEC is not formed in this work, the analysis in terms of effective temperature and chemical potential indicate that this is possibility.

This is an interesting idea. However, I am unsure whether enough new material is presented here. The paper seems to be more about reinterpreting results that have been published previously, which were described using different language, see e.g. [PRL 107, 107204 (2011)].

Reply:

We thank the Reviewer for a positive evaluation of the idea of our work. We believe that the Reviewer's concerns stem from the insufficient discussions in the original manuscript of the difference between the previously published and the present work. We agree with the Reviewer that even in our first work on the interaction of pure spin currents with the magnetization [PRL 107, 107204 (2011)], we recognized that the injection of spin current results in the excitation of different groups of incoherent magnons. Later, this qualitative understanding allowed us to achieve coherent spin-current driven auto-oscillations [Nature Mater. 11, 1028 (2012)] by utilizing selective magnon radiation losses. The systems studied in these works were designed to achieve efficient coherent precession, and were not suitable for the energy-resolved studies of magnon populations under the influence of spin current. As a result, none of the published results contain the information about the spectral distribution of magnons excited by spin current, which is needed for the analysis of thermodynamic properties of magnon gas in spin current-driven devices.

In our present work, we specifically designed an experimental system that allowed us to experimentally characterize the distribution of magnons excited by the pure spin current over a broad range of magnon energies, and confirm the theoretical predictions that the magnon gas can be driven by spin current into a quasi-equilibrium state that can be described thermodynamically. We believe that this is a significant step forward in the understanding of interaction of spin currents with magnons. In the revised manuscript, we have rewritten the third paragraph of the introduction to make this point clearer.

The Reviewer writes:

An important aspect is the connection between auto-oscillations and BEC. However, the sentence "Our findings provide support for the theoretically proposed interpretation of current-induced magnetization auto-oscillations in terms of the BEC of magnons" does not have a citation.

Reply:

We thank the Reviewer for pointing out this issue. We have added the proper citations in this sentence.

The Reviewer writes:

The only evidence given is that the critical currents are about the same. If this is a key result of the paper, then more convincing arguments should be presented. For example,

the authors have elsewhere reached auto-oscillation in similar devices, does that data agree with the new interpretation? This is important as, if at some later stage someone was to publish a paper saying that they have achieved BEC of spin Hall driven magnons, would that be a) an interesting result or b) a trivial one because it has been observed previously, but was described as auto-oscillations of the magnetization? This is not clear to me from reading the paper, and it is therefore difficult to evaluate its importance.

Reply:

We completely agree with the Reviewer that this issue deserves a separate discussion. We have added such a discussion on pages 8-9 of the revised manuscript.

As we have discussed above, the key result of our study is not a new interpretation of our previous results on the excitation of coherent dynamics, but the experimental information about magnon distribution in a broad energy interval, and the demonstration that the state of the magnon gas under the influence of spin current can be described by quasi-equilibrium thermodynamics. We have by no means claimed that coherent auto-oscillations observed in our previous works should be treated as Bose-Einstein condensation, since in all these works we used systems where the equilibrium was intentionally broken to prevent magnon-magnon interactions in the dense gas of driven magnons.

To address the Reviewer's comment, we now cite the paper by D. Snoke [Nature, 443, 403 (2006)] (Ref. 34 in the revised manuscript), where the criteria of BEC were discussed in the context of quasiparticle gases. According to these well-established criteria, to ascertain the formation of BEC, one should not only demonstrate the spontaneous emergence of coherence, but also show that the formation of coherent state is driven by the thermodynamics of quasiparticles. In this context, the opposite extreme is lasing, which results in the formation of a coherent state not associated with thermodynamic processes within the quasiparticle system. Thus, a demonstration of the magnon BEC driven by the pure spin current must include an experimental confirmation of the fact that the magnon gas can be described by the Bose-Einstein statistics. Our present work provides such an experimental demonstration for a simple model system. However, we also show that BEC does not occur in this system.

In our opinion, it remains an open question whether spontaneous emergence of coherence can be achieved in magnon systems driven by spin current, without breaking thermodynamic quasi-equilibrium. Such a demonstration could provide a substantial step forward in our understanding of interaction between magnon gases and spin current. We emphasize, however, that this can be an extremely challenging task. Therefore, our present results, which indicate its feasibility, should greatly stimulate the research of the thermodynamics of magnon gases.

The Reviewer writes:

Usually the anti-damping torque is written something like $M_x(M_x\sigma)$, in such a way that the magnetization effects the torque on itself. In this case, there is feedback of the magnetization dynamics into how the system is pumped. How does this usual form of the torque fit with the thermodynamic picture presented here, in which the pumping is consider to be simply broadband input of magnons? i.e. if there were coherent oscillations, would they be spontaneous due to condensation of magnons, or just from a feedback mechanism in the pumping?

Reply:

The Reviewer's comment raises an interesting question about the nature of magnon pumping by spin current. Unfortunately, there may be some confusion in the community stemming from the imprecise but commonly used terminology such as "injection of magnons", which creates the impression that spin current can create a "broadband input of magnons". Instead, it

provides a flow of angular momentum, which is converted into magnons by the spin system of the ferromagnet. As pointed out by the Reviewer, the Slonczewski's anti-damping torque in the LLG equation is generally believed to provide an adequate description for the effects on the magnetic system of the angular momentum flow, and it can be ultimately converted into the language of spin current-dependent magnon populations. However, this deterministic approach alone cannot describe the enhancement of fluctuations by spin-current in the subcritical regime (see [Phys. Rep. 673, 1 (2017)]). To account for these effects, one needs to include into the LLG-based model an additional source of fluctuations (e.g. Langevin's random field), whose thermodynamic characteristics largely determine the resulting magnon distribution modulated by the spin current. In other words, the thermodynamic description does not contradict the LLG approach mentioned by the Reviewer. In our opinion, it would be fruitful for the theoretical understanding of spin current-induced phenomena to combine these approaches. We believe that our present experimental results on the magnon distribution under the influence of the spin current should provide a foundation for such theoretical studies.

The Reviewer writes:

In Fig. 3 (c), the scale on the negative current data is confusing because it is labelled enhancement, even though it is a suppression for negative currents. Would it not make more sense just to label the y-axis as the ratio defined in the text?

Reply:

Following the Reviewer's suggestion, we have changed the axis label in Fig. 3.

The Reviewer writes:

In Fig. 4 the temperature/chemical potential are not plotted over the entire current range. If the fits to the ratio using the expression in the text give meaningful values, then they should be presented.

Reply:

Our results show that, at negative currents, the dominant effect of the spin current is the variation of the effective temperature, while at positive currents, the dominant effect is the change of the chemical potential. This does not indicate that the chemical potential is equal to zero at negative currents or that the effective temperature remains equal to the room temperature at positive currents. Instead, we can only ascertain that the variation of these parameters is too small to be reliably determined from the experimental data. To comply with the Reviewer's comment, we have added a discussion on page 8 of the revised manuscript to clarify this issue.

The Reviewer writes:

In conclusion, in its current form I would not recommend this paper for publication in Nature Communications. The claims that are made are ambiguous, and the paper essentially presents the reinterpretation of previously demonstrated experimental results. While this interpretation is interesting, it is a small step towards the more important results of either a) demonstrating spin current driven BEC or b) showing convincingly that auto-oscillations previously observed correspond to BEC.

Reply:

We hope that the clarifications provided in our response, and the changes in the revised manuscript, will convince the Reviewer that our claims are clear and by no means present a reinterpretation of the previous results. Although we did not achieve BEC in our experiments (and do not know if it is achievable in a system driven by spin current), we have clearly shown that the magnon gas subjected to the spin current can be described by the quasi-equilibrium Bose-

Einstein statistics, and determined its thermodynamic characteristics such as effective temperature and the chemical potential in a broad range of experimental parameters, which creates a substantial foundation for further theoretical and experimental studies. As an additional argument for the importance of these findings for a broad audience, we would like to mention that, very recently, the paper (Ref. 23), which much more indirectly addresses the issue of the chemical potential of magnon gases, was published in the Science Journal.

Reply to Reviewer #2

The Reviewer writes:

The manuscript by Dr. Demidov et al. presents BLS measurements on a Py/Pt nanostructure while a current is driven through the system.

The main claims of the paper are that i) the spin Hall effect in Pt injects a spin current into Py which affects the magnon population, and ii) the magnon population is affected in a way that is described by a Bose-Einstein distribution function with nonzero chemical potential and an effective temperature.

These main claims are novel in the sense that the BLS measurements while driving current, performed by the authors, present new results. Previous work (including in part that of the authors themselves) has demonstrated, either directly or indirectly, the existence of nonzero chemical potential, either by injecting spin current or by driving the system in other ways. The authors refer appropriately to the experimental achievements in this direction, although some more theory work in this direction could also be cited.

The paper will surely be of interest to researchers working in this field: this work presents a novel experiment that indicates that the magnon chemical potential is a sensible quantity in describing a quasi-equilibrium magnon system. The field of magnon spintronics/spin caloritronics/spintronics is recently shifting away from electron spin currents to magnon spin currents. The work by Demidov et al. provides important input on how to describe the latter spin currents. As mentioned above, it does not stand alone in the sense that there are now several indications that the magnon chemical potential is a useful quantity. Together with these other works (e.g. Refs.17,21 of the manuscript) I expect that this paper will be very influential.

Reply:

We thank the Reviewer for the positive evaluation of our work and for the appreciation of the novelty and the importance of our findings. We agree with the Reviewer that, in the original manuscript, we have omitted several important theoretical works indicating the importance of the chemical potential as a useful quantity for the description of quasi-equilibrium magnon systems. In the revised manuscript, we have added Refs. 19 and 20, and have rewritten the introduction to provide a better account of these works.

The Reviewer writes:

Regarding the validity of the claims, I have some concerns. That is, the claim rests on interpretation of the experimental data within the assumption that the main influence of the current is through the spin current in Pt. There are some aspects of this experimental system (i.e. a metallic magnet rather than an insulating one) and some aspects of the data that warrant questions:

***) To me, one of the most puzzling/intriguing aspects of the data is the asymmetry between the two directions of current. The authors do not discuss possible reasons for this in detail. In the linear response regime, one would expect effective temperature and chemical potential to be linear in the applied current, but I did not find an analysis that this is indeed the case – have the authors checked this? Could the authors provide more discussion of the asymmetry?**

Reply:

We completely agree with the Reviewer that the observed asymmetry between the current directions is one of the most intriguing aspects of our data. We also agree that, based on the general consideration, the current dependence of the effective temperature and the chemical potential should be continuous at $I=0$, which is also confirmed by the absence of discontinuity in the experimental dependence of the integral intensity on current (Fig. 2c). To address the Reviewer's comment, we have added a discussion on the asymmetry on page 8 of the revised manuscript. In particular, we now clearly indicate that, while the variation of the chemical potential is the dominant effect at positive currents and the variation of the effective temperature is the dominant effect at negative currents, both of these parameters are generally expected to vary for both current directions. We have also added Figure S3 in the supplementary material accompanying the manuscript, showing the analysis of the experimental data for small currents. This analysis shows that the extrapolation of the linear dependence of the chemical potential, observed at $I>0$, to $I<0$ does not provide a satisfactory description of the experimental data even at a modest $I=-5\text{mA}$, indicating that the chemical potential likely quickly saturates at $I<0$.

To further substantiate our experimental approach and analysis, we have performed measurements of the magnon population at different temperature values, attained by direct sample heating. The results presented in Figure S2 in the Supplementary Material clearly indicate that the variation of temperature results in a frequency-independent rescaling of the BLS intensity by T/T_0 , in agreement with the prediction of the Bose-Einstein statistics.

The Reviewer writes:

***) Since Py is a metal, there will be current flowing in it as well, and not only in the Pt – in fact, likely most of the current flows in the Py. Since the spin waves are not homogeneous in the direction of current, the current will influence the spin wave dispersion via the ordinary spin transfer torques (giving rise to what people call the spin wave Doppler effect). These will lower the $m=1$ mode for one direction of current, and raise it for the other one. Given the large current densities, it is not obvious to me that these effects may be ignored – have the authors included this effect in their analysis? Have the authors done a reference experiment without the Pt, e.g. replacing it with Cu?**

Reply:

We fully agree with the Reviewer. The utilization of thick Py necessary for the measurements of the magnon populations in a broad interval of frequencies results in a significant current shunting, which can modify the magnon dispersion spectrum. However, these modifications are negligible compared to the characteristic frequency scale in our measurements. Following the Reviewer's suggestion, we have indicated in the Methods section of the revised manuscript that the expected frequency shift due to the Doppler effect (new Ref. 35 in the revised manuscript) is of the order of 10 MHz, which is below the experimental resolution. In recent years, we have performed numerous BLS experiments aimed at the detection of the Doppler effect in Py. These experiments have clearly shown that this effect is too weak to be detected magneto-optically. To further illustrate the absence of the noticeable influence of the current

in Py on the dispersion spectra, we have included Figure S1 in the supplementary materials accompanying the revised manuscript, which shows the normalized BLS spectra recorded at $I=0$ and $I=-20$ mA. These data clearly show the absence of any noticeable modifications in the shape of the spectra under the influence of current.

The Reviewer writes:

***) If one estimates the change in total magnon density by estimating the injected spin and balancing it with relaxation, how does this then compare to the changes in the BLS data?**

Reply:

The dependence of the magnon density on current is expected to be hyperbolic (Ref. 26 in the revised manuscript), provided that the injection of spin can be described by the anti-damping torque included into the LLG equation. This should result in the linear dependence of the inverse BLS intensity on current, whose intercept represents the extrapolated critical current corresponding to the point of the complete damping compensation. Addressing the Reviewer's question, we have included Figure 4c in the revised manuscript, which shows that this is indeed the case, and added a corresponding discussion on page 9.

The Reviewer writes:

***) Are there Dzyalonshtinskii-Moriya interactions induced by the Py/Pt interface?**

Reply:

According to the published work on this subject, there should be Dzyalonshtinskii-Moriya interaction at the Py/Pt interface. However, since this interaction has an interfacial nature, it is known to have negligible influence on the magnon dispersion in Py films with the thickness exceeding 1-2 nm (see e.g. Nature Phys. 11, 825 (2015)). For the Py thickness of 10 nm used in our work, we expect the interfacial DMI contribution to be negligible.

The Reviewer writes:

***) The authors claim that the magnon gas is described by a Bose-Einstein distribution function with a nonzero chemical potential; strictly speaking they have only shown this for the low-energy tail of the distribution function.**

Reply:

We completely agree with the Reviewer that our data only show the effects of spin current on low-energy magnons, and that it is currently not possible to make a conclusion that the high-energy magnons are described by the same thermodynamic characteristics. However, as was shown in Ref. 7, the BEC can occur even in the absence of the complete thermalization between low- and high-energy magnons, because of the weak coupling between these two groups of magnons. To address the Reviewer's comment, we have added a short discussion on page 9 of the revised manuscript.

The Reviewer writes:

***) In line 91,92 the authors mention a long-standing debate – have they now resolved it?**

Reply:

We are grateful to the Reviewer for pointing out this issue. Now we see that we have forgotten to discuss it in the original manuscript. In fact, when the studied experimental system was designed, particular attention was given to the possibility to distinguish between the occupations of the ground magnon state corresponding to the minimum magnon energy, and the state corresponding to the quasi-uniform ferromagnetic resonance. Our data (Figs. 3b and 3c) clear-

ly show that the spin current results in the overpopulation of the former. Therefore, we can definitely say that the debate has now been resolved. The corresponding comment has been added on pages 6-7 of the revised manuscript. We now also show by the dashed lines in the revised Fig. 3b the frequencies corresponding to these two states.

The Reviewer writes:

The main criterion for publication for Nature Communications is that “In general, to be acceptable, a paper should represent an advance in understanding likely to influence thinking in the field.” Provided that the authors answer my questions above in a satisfactory way I believe that this paper will become part of a few works that demonstrate that the magnon chemical potential is important in describing magnon systems driven by spin current and as such will indeed influence the thinking of the field.

Reply:

We thank the Reviewer for the positive evaluation of our work. We hope that he/she will find our answers and the changes to the manuscript satisfactory and will recommend the publication of the revised manuscript in Nature Communications.

Reply to Reviewer #3

The Reviewer writes:

The manuscript by Demidov and co-workers addresses the phenomena through which a spin current may affect the magnon thermodynamics in a ferromagnetic material. The subject is novel, timely, and potentially interesting for the wide scientific community dedicated to Magnetism. The quality of the experimental data is very good and the manuscript provides strong evidence for its conclusions. To my opinion, the manuscript is fit for publication in Nature Communications.

Reply:

We thank the Reviewer for the positive evaluation of our work and for the recommendation to publish our manuscript in its present form.

Reviewers' comments:

Reviewer #1 (Remarks to the Author):

Having read the response to the comments, I still have reservations about this paper. My main concerns with are 1) that the paper presents reinterpretation of previously published results and 2) that the link to Bose-Einstein condensation is ambiguous and/or unsubstantiated. That the modified magnon distribution can be understood in terms of the Bose-Einstein distribution is an interesting result, but there are still parts of the paper that, to me, do not seem clear.

With regard to 1), the authors have stated that “none of the published results contain the information about the spectral distribution of magnons excited by spin current, which is needed for the analysis of thermodynamic properties of magnon gas in spin current-driven devices.” I am confused by this statement. When I look at Fig. 2 (a) in [Nature Mater. 11, 1028 (2012)], I see “information on the spectral distribution of magnons excited by spin current”. The graph is almost identical to Fig. 1 (a) in the current paper.

The authors state that the device in the current paper was specifically designed to characterise the distribution of magnons over a broad range of energies, different from in previous work. However, it is not made clear what these design differences are, and why this enables better characterisation.

With regard to 2), despite the added discussion, I think that this is still unclear. The added citations for the sentence “our finding provide support for the theoretically proposed interpretation of current-induced auto-oscillations in terms of BEC of magnons” do not contain any mention of the term “auto-oscillations”. These citations discuss both BEC and lasing, so it isn't clear which or both or either should be considered analogous to auto-oscillations. In the added discussion, the authors only talk about BEC, and not about lasing.

Further to this, I am still concerned about the feedback in the magnon injection process. While the authors agree that the injection process is complicate, they don't address this problem in the paper. In the response, they state that “the thermodynamic description does not contradict the LLG approach”. Regardless of the kind of modelling one uses, there is a clear difference between a system where there is input of incoherent quasiparticles (as the authors agree is not the case), and one in which there is feedback between the input and the dynamics of the system, as, for example, can be written down as the Slonczewski term in the LLG. This discussion has been made in the response, but not in the paper. In doing so it is possible that the authors will contribute to “some confusion in the community”.

Finally, I don't agree that because “the variation in the parameters is too small to be reliable” it

should not be plotted – the reader should be able to see that this is the case if the error bars signify that the uncertainty is large.

For these reasons, I would still not recommend this manuscript for publication in Nature Communications.

Reviewer #2 (Remarks to the Author):

I have reviewed the reply of the authors to the criticism of myself and the other reviewers. In my opinion, the authors have adequately responded to all criticism and have modified the manuscript appropriately. Hence, I now recommend publication of the manuscript.

Reply to Reviewer #1

The Reviewer writes:

Having read the response to the comments, I still have reservations about this paper. My main concerns with are 1) that the paper presents reinterpretation of previously published results and 2) that the link to Bose-Einstein condensation is ambiguous and/or unsubstantiated. That the modified magnon distribution can be understood in terms of the Bose-Einstein distribution is an interesting result, but there are still parts of the paper that, to me, do not seem clear.

Reply:

We thank the Referee for acknowledging that our results are interesting. We hope that further clarifications given below and the changes made in the manuscript will resolve all the Reviewer's concerns.

The Reviewer writes:

With regard to 1), the authors have stated that “none of the published results contain the information about the spectral distribution of magnons excited by spin current, which is needed for the analysis of thermodynamic properties of magnon gas in spin current-driven devices.” I am confused by this statement. When I look at Fig. 2 (a) in [Nature Mater. 11, 1028 (2012)], I see “information on the spectral distribution of magnons excited by spin current”. The graph is almost identical to Fig. 1 (a) in the current paper.

Reply:

The spectral peaks shown in Fig. 2(a) of our earlier paper cited by the Reviewer do not provide spectroscopic information needed for the analysis of spectral distribution, for the following reasons. The BLS spectra represent a convolution between the spectral resolution of the measurement technique and the spectral magnon distribution weighted by the frequency-dependent sensitivity of the technique. The frequency resolution in the cited measurement was comparable to the width of the peaks shown in Fig. 2(a) of the cited paper, while the accessible spectral range was significantly smaller than in our present work, because of the smaller thickness of the magnetic film resulting in a flatter dispersion. As a result of a combination of limited spectral resolution and spectral range, the spectral peak in Fig. 2(a) of the paper cited by the Reviewer is a distorted Gaussian dominated by the instrumental broadening, containing little information about the spectral distribution.

To determine the chemical potential and the effective temperature, one needs to experimentally access the spectral distribution of magnons over a range of frequencies comparable to the frequency of the lowest magnon state, which could not be realized in the previous studies of devices based on 5 nm-thick or even thinner magnetic films. Moreover, because of the nearly flat dispersion for thin films, it was previously not possible to experimentally distinguish the quasi-uniform magnon mode from the lowest-energy mode, which is essential for the ability to distinguish BEC from lasing (see also our answer to the fourth question). By contrast, in our present experiment based on a thicker magnetic wire, we are able to unambiguously conclude that the magnon state overpopulated due to the injection of pure spin current is the lowest-energy state. Finally, in all the spin-Hall oscillator devices studied up to now, the magneti-

zation was not spatially uniform due to the geometrical effects and the non-uniform Oersted fields of the driving current, resulting in a complex dispersion spectrum (usually including localized modes) that strongly varied with the driving current. All of these features make the previously studied systems unsuitable for the analysis of the magnon populations under the influence of spin current.

We have included a detailed discussion in the Methods section of the revised manuscript to make these points clearer.

The Reviewer writes:

The authors state that the device in the current paper was specifically designed to characterise the distribution of magnons over a broad range of energies, different from in previous work. However, it is not made clear what these design differences are, and why this enables better characterisation.

Reply:

To overcome the difficulties listed in our previous response, we base our experimental system on a relatively thick Py film patterned into a narrow wire. The thickness of 10 nm was chosen as a compromise between two opposite trends. On the one hand, our experimental technique is predominantly sensitive to spectral states with wavevectors smaller than the inverse size of the probing spot. The studied relatively thick film exhibits a sufficiently steep magnon dispersion, providing experimental access to a sufficiently broad spectral range. On the other hand, the effects of spin current scale inversely with the film thickness. The film is sufficiently thin to ensure that these effects remain significant, allowing us to achieve chemical potential approaching the lowest-energy magnon state. Additionally, the Py film is shaped as a narrow wire with the magnetization directed along its axis. The quantization of magnon spectrum in the transverse direction removes the frequency degeneracy of the low-energy magnon modes, enabling us to clearly experimentally distinguish quasi-uniform mode from the lowest-frequency mode. Moreover, the large aspect ratio of the Py wire ensures that the internal static magnetic field and the static magnetization are uniform in the active device area, avoiding magnon edge states that could complicate the magnon spectrum, and allowing measurements at moderate static fields at which the accessible spectral range is comparable to the frequency of the lowest magnon state. The geometry of the sample is also optimized to produce a uniform Oersted field of the driving current in the active device area, which results in a negligible current-induced modification of the magnon spectrum, aside from its uniform shift discussed in the text.

We included a detailed discussion in the Methods section of the revised manuscript to make this clearer.

The Reviewer writes:

With regard to 2), despite the added discussion, I think that this is still unclear. The added citations for the sentence “our finding provide support for the theoretically proposed interpretation of current-induced auto-oscillations in terms of BEC of magnons” do not contain any mention of the term “auto-oscillations”. These citations discuss both BEC and lasing, so it isn’t clear which or both or either should be considered analogous to auto-oscillations. In the added discussion, the authors only talk about BEC, and not about lasing.

Reply:

Both lasing and BEC are associated with spontaneous formation of a macroscopic coherent state of a bosonic system, manifested in case of magnons as coherent magnetization auto-oscillations. The difference between these two cases is in the state of the magnon gas. While lasing is a strongly non-equilibrium effect not associated with a quasi-equilibrium distribution of magnons, BEC is associated with a quasi-equilibrium state of magnon gas that can be described thermodynamically. Lasing can occur in any spectral state (determined by the relevant selection rules), not necessarily the lowest-energy state of the dynamical system. In contrast, BEC always occurs in the lowest-energy state, while the populations of all the other states are simply determined by their energies relative to the energy of this state. Early work on spin current-induced effects, in particular the famous predictions by L. Berger, focused on the first scenario. It is also widely believed that STT-driven auto-oscillations usually involve quasi-uniform mode of the system, which for in-plane magnetized systems is not the lowest-energy mode.

Up to now, there has been no direct experimental evidence allowing one to distinguish between the two scenarios predicted in the cited theoretical works. Thanks to the specially designed experimental system, we are able to distinguish between them and confirm that the magnon state overpopulated due to the injection of the pure spin current is, indeed, the lowest-energy state. Moreover, our results strongly support the BEC scenario, since:

- a) We directly show that the magnons driven by spin current form a thermodynamic distribution with certain effective temperature and chemical potential;
- b) We show that the chemical potential increases with increasing spin current and approaches the lowest -energy magnon state, as expected for the BEC scenario.

We emphasize that the observations summarized in a) and b) allow us to qualitatively distinguish between BEC and lasing.

To address the Reviewer's question, we have extended the discussion on page 9 of the revised manuscript, to clearly explain the difference between lasing and BEC. We have also modified accordingly the phrase on page 11 mentioned by the Reviewer.

The Reviewer writes:

Further to this, I am still concerned about the feedback in the magnon injection process. While the authors agree that the injection process is complicate, they don't address this problem in the paper. In the response, they state that "the thermodynamic description does not contradict the LLG approach". Regardless of the kind of modelling one uses, there is a clear difference between a system where there is input of incoherent quasiparticles (as the authors agree is not the case), and one in which there is feedback between the input and the dynamics of the system, as, for example, can be written down as the Slonczewski term in the LLG. This discussion has been made in the response, but not in the paper. In doing so it is possible that the authors will contribute to "some confusion in the community".

Reply:

We are pleased that the Reviewer agrees with our vision of the magnon injection process. Following the Reviewer's suggestion, we have included the corresponding discussion on pages 10-11 of the revised manuscript. We now explicitly indicate that there is a feedback in the magnon injection process.

The Reviewer writes:

Finally, I don't agree that because "the variation in the parameters is too small to be reliable" it should not be plotted – the reader should be able to see that this is the case if the error bars signify that the uncertainty is large.

Reply:

Complying with the Reviewer request, we have added in Fig. 4 the experimental data for the chemical potential at $I < 0$ and for the temperature at $I > 0$, and the relevant error bars. We have also modified accordingly the discussion of the data on pages 7-8 of the revised manuscript and indicated the values and the standard errors for both fitting parameters on page 7.

The Reviewer writes:

For these reasons, I would still not recommend this manuscript for publication in Nature Communications.

Reply:

As described above, we have complied with all the Reviewer's requests. We hope that he/she will recommend the publication of the revised manuscript in Nature Communications.

Reviewers' Comments:

Reviewer #1 (Remarks to the Author):

Having read the modified manuscript and authors replies to comments, I believe that the authors have addressed all of my concerns. I am therefore happy to recommend publication in Nature Communications.

Reply to Reviewer #1

The Reviewer writes:

Having read the modified manuscript and authors replies to comments, I believe that the authors have addressed all of my concerns. I am therefore happy to recommend publication in Nature Communications.

Reply:

We thank the Reviewer for the recommendation to publish the revised manuscript in Nature Communications.